# Polyphenols in Cereals: State of the Art of Available Information and Its Potential Use in Epidemiological Studies

**DOI:** 10.3390/nu16132155

**Published:** 2024-07-06

**Authors:** Donatella Bianca Maria Ficco, Katia Petroni, Lorenza Mistura, Laura D’Addezio

**Affiliations:** 1Consiglio per la Ricerca in Agricoltura e l’Analisi dell’Economia Agraria (CREA)—Centro di Ricerca Cerealicoltura e Colture Industriali, S.S. 673 m 25200, 71122 Foggia, Italy; 2Dipartimento di Bioscienze, Università degli Studi di Milano, Via Celoria, 26, 20133 Milan, Italy; katia.petroni@unimi.it; 3Consiglio per la Ricerca in Agricoltura e l’Analisi dell’Economia Agraria (CREA)—Centro di Ricerca Alimenti e Nutrizione, Via Ardeatina 546, 00178 Roma, Italy; lorenza.mistura@crea.gov.it (L.M.); laura.daddezio@crea.gov.it (L.D.)

**Keywords:** antioxidants, cereals/pigmented cereals, composition databases, polyphenol intake, observational studies

## Abstract

Cereals are the basis of much of the world’s daily diet. Recently, there has been considerable interest in the beneficial properties of wholegrains due to their content of phytochemicals, particularly polyphenols. Despite this, the existing data on polyphenolic composition of cereal-based foods reported in the most comprehensive databases are still not updated. Many cereal-based foods and phenolic compounds are missing, including pigmented ones. Observational epidemiological studies reporting the intake of polyphenols from cereals are limited and inconsistent, although experimental studies suggest a protective role for dietary polyphenols against cardiovascular disease, diabetes, and cancer. Estimating polyphenol intake is complex because of the large number of compounds present in foods and the many factors that affect their levels, such as plant variety, harvest season, food processing and cooking, making it difficult matching consumption data with data on food composition. Further, it should be taken into account that food composition tables and consumed foods are categorized in different ways. The present work provides an overview of the available data on polyphenols content reported in several existing databases, in terms of presence, missing and no data, and discusses the strengths and weaknesses of methods for assessing cereal polyphenol consumption. Furthermore, this review suggests a greater need for the inclusion of most up-to-date cereal food composition data and for the harmonization of standardized procedures in collecting cereal-based food data and adequate assessment tools for dietary intake.

## 1. Introduction

A substantial number of meta-analyses of observational studies and randomized controlled trials have provided robust evidence that a higher adherence to the Mediterranean diet, characterized by a high consumption of fruits, vegetables, wholegrain cereals, legumes, nuts and seeds, olive oil, a moderate wine consumption and low intake of animal-derived products, is associated with a reduced risk of chronic degenerative diseases and total mortality [1]. Among foods included in the Mediterranean diet, wholegrain cereals have been associated with a reduced risk of cardiovascular diseases, total cancer, type 2 diabetes, obesity, and all-cause mortality in several large-scale prospective cohort studies, providing support for the current recommendation of increasing wholegrain consumption as part of a healthy diet for the prevention of chronic diseases [2,3]. Cereals are an important source of energy and nutrients in the human diet and are one of the staple foods in the world [4,5].

Total world cereal production in 2022 was estimated at 3059 million tonnes. Estimates of domestic supplies for 2021 were highest in Asia (1663.2), followed by North America (393.32), Africa (316.38) and the European Union (281.10) (https://www.fao.org/faostat/, accessed on 30 May 2024). Since 2010, there has been a trend towards an increase in these figures. In the European Union, wheat and its products account for the largest share of the cereals available for domestic consumption (115.66 million tonnes), followed by maize (66.11) and barley (46.26) (https://www.fao.org/faostat/, accessed on 30 May 2024).

The EU consumption of cereals in 2024/25 is not expected to change substantially [6], with a daily mean intake of cereals for the adult population of 219 g publicly available through the EFSA Comprehensive Food Consumption Database website (https://www.efsa.europa.eu/en/data-report/food-consumption-data, accessed on 30 May 2024). Cereals are the edible seeds of grasses from the *Poaceae* family and include bread and durum wheats, oat, barley, rice, sorghum, millet, maize and rye. Cereal grains consist of three major components: 65–75% carbohydrates, mainly starches, 6–15% proteins, 2–6.5% lipids, and other minor, albeit important, nutrients and non-nutrients, such as carotenoids, minerals, dietary fibers, vitamins, and bioactive compounds [7].

Among non-nutrients, polyphenols are secondary metabolites naturally produced by plants under physiological developmental cues as well as in response to abiotic and biotic stresses and are more accumulated in the outer layers of the grains [8,9]. Cereals are an important source of polyphenols with potential health benefits [10]. Recently, extensive literature has been reported on pigmented cereals for their many phytochemicals, including anthocyanins and carotenoids associated with numerous health benefits [11,12].

Research evidence indicates that the beneficial properties of wholegrain cereals result from their unique composition in phytochemicals, including bioactive compounds present mainly in cereal bran and germ that are usually removed during refining [13,14,15].

The current review provides an up-to-date critical survey of existing food composition and non-nutrient databases with a specific focus on classification and inclusion therein of polyphenols from cereals. As for food consumption data, the EFSA European Comprehensive Food Consumption Database (CFCD) [16] and the FAO/WHO Global Individual Food Consumption Data Tool (FAO/WHO GIFT) [17] were explored. The former provides data from most European countries, while the latter collects dietary data from all regions of the world, focusing on low- and middle-income countries. The state-of-the-art of databases are reported at the date of the drafting of the present manuscript (May 2024). An analysis of dietary assessment methods to measure nutrient intake in epidemiological studies is also reported, followed by a perspective to overcome evidenced limits.

## 2. Defining Polyphenols: Structure and Properties

Polyphenols, also named phenolics or phenolic compounds, are natural compounds of the secondary metabolism synthesized in plant cells via the phenylpropanoid pathway. The key enzymes involved in the biosynthesis of the various classes of polyphenols are reported in Figure 1.

Polyphenols are widely studied for their human health-related benefits against oxidative stress and chronic diseases, such as inflammatory bowel diseases, cardiovascular disease, cancer and obesity [18]. The different classes of polyphenols have unique chemical and biological properties, depending on their structure, which can be characterized by one or multiple aromatic (phenol) rings with one or more hydroxyl substituents. The most common polyphenols found in cereals include phenolic acids, classified as hydroxybenzoic or hydroxycinnamic acids, flavonoids and lignans.

Phenolic acids are the main compounds present in cereal grains, acting as building materials for the cell wall structures. The presence of a single phenol ring along with a one-carbon (C6–C1) or three-carbon (C6–C3) side chain is a distinguishing feature of hydroxybenzoic and hydroxycinnamic acids, respectively. The hydroxybenzoic acids present in cereals are mainly represented by p-hydroxybenzoic, protocatechuic, vanillic, syringic and gallic acids, while the hydroxycinnamic acids by p-coumaric, caffeic, ferulic and sinapic acids. Ferulic acid is the most abundant phenolic acid in cereals, accounting for up to 90% of total phenolic acids and is mainly concentrated in the external layers of kernel [19]. The different substituents then influence their antioxidant capacity and bioactivity.

Flavonoids consist of a three-ring structure in the C6–C3–C6 form, with different substituents, such as hydroxyl and methoxyl groups, characterizing different classes of flavonoids, such as flavones, flavanones, flavonols, flavanonols, flavanols and anthocyanidins.

Finally, lignans belong to the group of the diphenolic compounds derived from the combination of two phenylpropanoid C6–C3 units. Phenolic compounds can also be distinguished into free, esterified and insoluble-bound forms, for their ability to form conjugates with other compounds such as fatty acids (i.e., soluble esters) or to form insoluble macromolecules (i.e., insoluble-bound phenolics) [20]. Insoluble-bound phenolics are extensively found in the cell wall and mainly in the bran of cereals [21]. Overall, the amount and specific polyphenolic profile of cereal species and varieties may depend on the plant genetic background, the environmental growth conditions, including the possible occurrence of abiotic and biotic stresses, as well as the agronomic practices applied [9]. Furthermore, different varieties of pigmented cereals, adapted to local pedoclimatic conditions, are cultivated all around the world. As an example, pigmented corn is mainly cultivated in Central and South America (i.e., Mexico, Peru and Bolivia), but many other varieties considered promising functional foods are also cultivated in other countries (i.e., Russia, Turkey, Europe, Thailand) [22,23]. The polyphenol composition of many of these varieties has been determined, indicating the existence of specificities in anthocyanin glycosylation and acylation, and of differences in hydroxycinnamic acid composition that contribute to the complexity of polyphenol characterization of these geographical accessions (for a review see Colombo et al. [22]). A similar complexity has been reviewed for pigmented rice and wheat varieties [24,25].

## 3. State of the Art of Available Databases on Polyphenol Contents in Cereals

Despite the importance of cereals in human health and the widespread consumption of several types of cereal-based products, the information about polyphenols composition and related biological effects of cereal foods deposited onto web-based databases is limited [8,26] (Table 1). 

More specifically, the EuroFIR Bioactive Substances in the Food Information Systems (eBASIS) database (updated to 2016 within the BACCHUS project) for bioactive intakes in Europe includes 10,599 records, 86 plants, and covers 242 individual compounds, including anthocyanins, ellagitannins and ellagic acid, flavanols, flavanones, flavones, flavonols, and proanthocyanidins Plumb et al. 2017 [26] and Pounis et al. 2016 [27]. However, when searching for the plant family ‘*Poaceae* (*Graminaceae*)’ and selecting for each cereal plant name (i.e., wheat, barley, rice, etc.), information about sampling, processing, analytical methods on the composition of polyphenol constituents is incomplete. For instance, when searching for ‘Extractable polyphenols (EPP)’ and selecting ‘wheat’, only some information, such as the ID, reference, number of replicates, standard deviation, and the analytical method, mainly measured as total content (i.e., spectrophotometric method), were reported. Similarly, when searching the USDA Global Branded Food Products Database (version 3.2/2015), which is the major source of food composition data in the United States, the flavonoid composition of some cereal grains was found. In particular, flavanols, anthocyanidins and flavanones/flavones were identified in barley, purple wheat and sorghum, respectively (Appendix A).

Finally, Phenol-Explorer, a publicly available web-based database (version 3.6/2015), is the first attempt to systematically collect data on polyphenols in foods, including the content and composition not only in cereal grains, but also after processing and/or cooking (Appendix A). When searching for ‘cereals and cereal-based products’, 23 items, including barley, wheat, maize, rice and rye, were found. The database provides information on the loss or gain of each polyphenol compound after the processing procedures (i.e., retention factor). However, despite the amount of information, not all data traceable back to the published literature have been reviewed or updated, thus possibly impacting the accuracy needed for estimating polyphenol intake from cereals in epidemiological studies. Based on the analysis of economic data from January to June 2023, the ISMEA—Institute of Services for the Agricultural and Food Market—evidenced a significant increase in Italian household spending in cereal derivatives (+15.6%, on average), where the driving forces were ‘breakfast products’ (+18%) and ‘bread and substitutes’ (+17.8%). Further increases in cereal consumption were found in rice (+26%) and pasta (+11%), while wheat flour remained below average (+5.8%) (ISMEA processing on NielsenIQ data, domestic purchases 3/2023). Despite the ever-increasing demand for cereals, some categories of cereal-based foods, such as snacks, precooked frozen and infant cereal foods, and novel formulations of pasta and bread with the addition of proteins, dietary fibers, phytochemicals from other sources, are not represented in databases (Table 1). This gap is even more pronounced for pigmented cereals, whose purple, red, blue and black pigmentation is linked to polyphenols, such as anthocyanins, proanthocyanidins, flavonols, phenolic acids, and lignins [28,29,30], with a role in reducing the risk of chronic diseases, like hypertension, heart diseases, cancers, diabetes and obesity due their antioxidant properties [31].

The food and nutrient databases in the current forms do not provide truly comprehensive food composition data [32]. This is even more evident in the case of cereals and particularly of pigmented cereals. Furthermore, the polyphenol characterization depends on species and cultivars, climate, agronomic practices, post-harvest and food processing, methods of domestic preparation of cereal-based foods. This complexity requires validated and rigorous analytical methods that enable a specific and reliable identification of single types of polyphenols, as evidenced by Schroeter et al. [33] and Yeung [34] for flavanols and procyanidins. Despite certain analytical methods and procedures being widely used (e.g., HPLC, GC-MS), standard analytical methods for the determination of polyphenol composition have not been established for all polyphenols classes. To this aim, accredited analytical standard methods (i.e., validated ICC, AOACI, ISO, etc.) should be adopted, in which sample collection and preparation, detection systems, and identification are precisely described and standardized. These standardized procedures will overcome those limitations that currently hinder cross-study comparisons, meta-analyses, and multi-source food intake data evaluations. With these technical improvements, the ability to integrate information from different data sources will be possible and will allow to establish a definitive understanding of what is currently known and what is missing in order to make food composition data more complete and provide *findability, accessibility, interoperability, and reusability (FAIR)*.

The implementation of data, encompassing both food and related occurrences in cereal genetic sources, including pigmented varieties, is fundamental for estimating polyphenol intake and determining their health-promoting properties. Also, information on processing procedures needs to be included in databases, since cereals have to be somehow processed (i.e., milling, preparation, baking, cooking, etc.) in order to be consumed. During processing, a general reduction in nutrients and polyphenols can be observed, which could eventually be counterbalanced by an increase in digestibility and bioavailability [15,35,36]. In general, the molecular structure of polyphenols determines their bioavailability, but also the type of food matrix may significantly affect their absorption. As an example, acylated anthocyanins are less bioavailable than non-acylated [37]. Glycosylated anthocyanins are more stable and water soluble than their aglycone counterparts, but their bioavailability results limited since they can only be absorbed through glucose transporters, such as SGLT [38]. The overall bioavailability of ^13^C-labelled cyanidin 3-glucoside (C3G) has been in fact estimated to be 12%, by measuring plasmatic levels of both C3G, its phase I and phase II metabolites, and microbiota degradation products [39,40]. Nonetheless, polyphenols need to be released from the food matrix in order to be absorbed and transported via the bloodstream to the target tissue, in order to display their biological activity. Some studies have highlighted that flavonoids can bind the food matrix through covalent or non-covalent bonds, that they can influence nutrient absorption and in turn be influenced by nutrients in their bioavailability and biological activity (for a review see Zhang et al. [41]). As an example, wheat proteins may interact with flavonoids forming indigestible complexes that reduce their antioxidant activity [42]. However, the addition of citric acid in food preparations and supplements in order to avoid flavonoids oxidation was found to enhance their bioavailability by releasing them from the food matrix of pigmented maize [43]. On the other hand, despite the thermal treatment of pigmented maize flour reduced the anthocyanin content to some extent, the high temperature applied during cooking processes had the advantage to increase their bioavailability by releasing them from the food matrix [44]. More studies are, however, needed to understand the influence of cereal as well as other plant food matrices on both the bioaccessibility and bioavailability of polyphenols [41].

## 4. Focus on Pigmented Cereals for Foods and Beverages

Increasing evidence has shown the potential for anthocyanin-rich cereal-based products in preventing degenerative chronic diseases. Although the anthocyanin content in pigmented cereals is generally lower compared to that in fruits and vegetables [15], some newly developed cereal varieties exhibit anthocyanin levels close to those of some grapes and berries [45,46,47]. Furthermore, the daily demand for pigmented cereals is likely to increase due to the growing consumers’ shift to a healthy diet and lifestyle, becoming an important factor in consumers’ choice [48,49,50,51,52].

Compared with non-pigmented cereals, pigmented cereals contain a high concentration of proanthocyanidins (i.e., red rice), phlobaphenes (i.e., red corn) and anthocyanins in rice, wheat and corn [29,37]. Concerning anthocyanins, black and purple rice contain high amounts of cyanidin 3-glucoside [24], whereas blue and purple corn contain cyanidin 3-glucoside, peonidin 3-glucoside, pelargonidin 3-glucoside and malonylated derivatives in the aleurone or in the pericarp of kernels, respectively [46]. Finally, blue-aleurone wheats contain delphinidin 3-glucoside, delphinidin 3-rutinoside and malvidin 3-glucoside while in purple-pericarp wheat and durum wheat cyanidin 3-glucoside, and/or peonidin malonylglucoside, peonidin 3-galactoside and malvidin 3-glucoside have been observed [28,30]. These compounds have been associated with high antioxidant capacity and anti-inflammatory activities [53]. Pigmented cereals are treated as suitable ingredients as they provide functional attributes apart from color (for a review, see Bassolino et al., [54]). Recently, there has been evidence of the expansion of a niche market for pigmented rice, represented by black, purple and red rice varieties, consumed as wholegrains for their high levels of phenolic compounds, particularly anthocyanins, located in the external layers of seeds [55]. Examples of typical pigmented cereal-based foods and beverages are Tapai, a black rice fermented food of Indonesia and Tapuy, a Philippine wine made from Ballatinao black rice and a traditional starter culture (Bubod). Italy is the main rice producer in Europe, particularly in the area between the Piedmont and Lombardy regions, where a large collection of registered varieties characterized by different nutritional and technological features is maintained. The main popular registered varieties of black rice grown in Italy, and in particular in the Piedmont region, are Artemide, Venere and Nerone, while those of red rice are represented by Ermes and Ris Rus with a great culinary appreciation by the Italian population [56]. Additionally, the Italian brand “Riso Scotti” has created an all-Italian value chain, marketing the versatile black rice Venere, suited for preparation of “risotto”, pasta, cous cous, cakes and other snacks.

Apart from being rich in antioxidants, pigmented rice varieties have lower digestibility than conventional rice, since they are characterized by differences in gelatinization behavior and starch structure, so that they can be used in calorie-restricted diets for weight loss beyond that in gluten-free foods for celiac patients [57]. Scientific studies have attributed several health benefits to pigmented rice, such as antioxidant, antidiabetic and anticancer properties [24]. Some traditional pigmented rice varieties are used for medications in Ayurveda traditional medicine and as functional foods for promoting lactation, such as red rice Rakthashali from Kerala, India, or as beneficial foods to ameliorate blood circulation, gastritis and peptic ulcers, such as black rice Kavuni from Tamil Nadu, India [24,58,59]. In a recent systematic review via a meta-analysis study, researchers found that diets based on anthocyanin-rich fruits and vegetables, including black rice and purple and black wheats, positively impacted the gut microbiota [60].

In Perú, an example of daily beverages made from purple maize is represented by chicha morada, a dark non-alcoholic drink that originated in the Andes and spread throughout Latin America [61]. Upon cooking and the addition of dried or fresh fruits to chicha morada, an appreciated dessert, mazamorra morada, can be also obtained [62]. Other maize-based foods are tortillas, tortilla chips and other Mexican foods produced using masa, prepared through the nixtamalization of blue maize, an alkali thermal treatment producing flour [63]. Typical Mexican biscuits named polvorones are also produced from pigmented corn instead of regular wheat flour, resulting in better color, flavor and acceptability by consumers [64]. In northern regions of Italy and Spain (Galicia), ancient maize varieties cultivated in mountainous areas are often characterized by the presence of pigments, such as phlobaphenes, anthocyanins, flavonols and phenolic acids in “Nero Spinoso” [65] and “Millo Corvo” [66], and phlobaphenes in “Rostrato Rosso di Rovetta” [67]. Anthocyanin-rich popcorn, polenta and sweet corn adapted to the European photoperiod were developed through recurrent backcrosses with a purple synthetic variety and have been tested as suitable for a gluten-free diet for individuals affected by celiac disease [46,47,68,69]. Similar results were obtained on pigmented maize varieties from Turkey [70].

The Italian dishes of cooked rice such as “risotto” and cooked corn flour such as “polenta” are the best cooking methods to preserve anthocyanin and polyphenol content. This contrasts with other cereal-based products like pasta, where boiling may leach polyphenols into water, which is then discarded, resulting in decreased polyphenol content. In this latter case, the selection of wheat/durum wheat cultivars high in polyphenols by breeding approaches becomes even more important. Moreover, bread and durum wheats, including pigmented cultivars, need to be milled into flour before being used for the production of naturally colored pasta, bread and other cereal-based products [71,72,73]. Recently, due to the localization of phenolics in the bran layers, a careful selection of the most appropriate fractionation process (debranning/micronization and air-classification) has been considered in order to preserve the wheat bran layers richest in phytochemicals (i.e., the anthocyanin-rich fractions in pigmented wheats) [43] and incorporate them into fortified pasta, snacks and bakery products [74,75].

Some pigmented landraces of bread and durum wheat, originally from Ethiopia, have been cultivated in Italy as regionally produced niche crops in order to satisfy the increasing consumers’ demand for novel products richer in health-promoting ingredients. For instance, the company Granomischio maintained and cultivated in the Apulia region, in the areas of Daunia Mountains, some purple and red durum wheat landraces with interesting features for the niche market of traditional cereal-based foods [76]. Due to the lower yield potential and quality of pigmented ancient wheats compared to modern yellow cultivars, these genetic materials being important sources of anthocyanins have been used in breeding programs to improve the nutritional and bioactive value of new cultivars.

Several preclinical studies demonstrated that supplementation with pigmented rice or wheat prevent metabolic syndrome by ameliorating the negative effects of diet-induced obesity and provide a neuroprotective effect in animal models of neurodegenerative diseases, such as Alzheimer’s and Parkinson’s diseases (for a review, see Bassolino et al. [54]). Similarly, the preventive effect of pigmented corn has been ascertained on a variety of animal models of chronic diseases, such as cardiovascular disease, obesity, diabetes, cancer, and some genetic diseases [23,54]. However, only a very limited number of human intervention studies have been undertaken, some of which considered the beneficial effect of cereal-based extracts as dietary supplements.

Regarding antioxidant potential, some authors provided results on the health effects of cereal-based rich-anthocyanins purified extracts, showing protection of the vascular endothelium, lowering blood pressure, and an improvement in low-density lipoprotein (LDL) oxidation (for a review, see Francavilla and Joye et al. [15]). Recently, healthy characteristics of beer by using pigmented cereals as raw ingredients have been considered [31]. Clinical trials showed that a moderate consumption of beer could mimic some of the previously reported health properties of red wine, improving the lipid profile and reducing inflammatory biomarkers related to atherosclerotic process [77]. By analyzing phenolics-rich extracts of different cultivars of pigmented corn and wheat and of selected debranning fraction, Parizad et al. [78] found an inhibition of carbohydrate metabolism enzymes (i.e., pancreatic α-amylase and intestinal α-glucosidase) and an anti-inflammatory activity in Caco-2 cells transiently transfected with a luciferase reporter gene responding to cytokine stimulation. The authors suggest that these effects may not be attributed to a specific component in the extracts but rather to a synergy among yet unidentified components, suggesting the use of these fractions as possible additives in dairy foods.

An anthocyanin-rich extract from purple corn has been demonstrated to reduce obesity-associated low-grade inflammation and inflammatory trigeminal pain to an extent similar to salicylic acid [79,80]. Based on these preclinical studies, a pilot intervention study has been undertaken including a daily supplementation of purple corn extract (RED) to Crohn’s disease patients under Infliximab therapy, a monoclonal antibody against TNF-α. The RED extract showed an optimal safety profile, and a significant reduction in TNF-α and other pro-inflammatory cytokines/chemokines, suggesting its potential use as adjuvant therapy to reduce symptoms and prevent relapses in Crohn’s disease patients [81,82].

Recently, a randomized controlled human trial has been conducted to investigate the effect of pigmented rice consumption on cardiometabolic risk, showing significant beneficial effects on glucose levels by delaying carbohydrate absorption through the inhibition of α-amylase and α-glucosidase, as well as positive effects on weight and diastolic blood pressure [83].

## 5. Estimating Polyphenol Intake from Cereals: Challenges and Opportunities

### 5.1. Observational Studies on Polyphenol Intake

Despite global cereal consumption and potential health benefits, there are few observational studies on cereal polyphenol intake and its association with disease outcomes in human populations. While evidence from experimental studies supports a protective role for dietary polyphenols against cardiovascular disease, diabetes and cancer, observational epidemiological studies are limited and inconsistent [83].

A number of studies have been carried out on the dietary intake of polyphenols. Zamora-Ros et al. [84] analyzed data from the EPIC cohort study conducted in 10 European countries showing that cereals are a non-negligible source of total polyphenols and the main source of lignans in non-Mediterranean countries. After coffee, breads and cereals were the second most important source of phenolic acids and total polyphenols in Finnish adults [85], while for the French adult population, cereals are the sixth largest source of polyphenol intake (46 mg/d compared to 658 mg/d of non-alcoholic beverages, mainly from coffee, wine and apple juice). Main contributors within the cereal group were refined wheat-flour products (66%), wholegrain wheat-flour products (26%), breakfast cereals (7%), and rice (2%) [86]. In Polish adults, wheat flour products were the main dietary source of flavones, dark bread was one of the main sources of lignans, and cereals, dark bread and pasta were reported as the main sources of alkylphenols [87]. More recently, cereals were estimated to be the main source of total polyphenols in the adult Danish population, after non-alcoholic beverages (coffee accounted for more than 55%) and cocoa products [88]. Vingrys et al. [10] estimated the dietary intake of polyphenols from cereal foods in a large sample of adults from Melbourne, Australia, and reported an intake of 86.9 mg/day. Phenolic acids were the most commonly consumed compounds and are mainly associated with wheat consumption, which was commonly consumed by the study participants [89].

Estimating polyphenol intake is challenging due to the wide variety of compounds found in foods and the factors that can affect their levels, such as plant variety, season of harvest, and how the food is processed and cooked, as evidenced in the work of Zamora-Ros et al. [83] reviewed several observational epidemiological studies on the relationship between polyphenol intake and cancer applied diverse estimation methods. Zamora-Ros et al. [83] reached the conclusion that the focus should be on estimating the intake of individual polyphenols, rather than considering them collectively, to understand the role of polyphenols in disease prevention. This underlines the importance of the assessment of intakes from individual food sources and the need for detailed information on the content of these compounds in a wide variety of cereals and cereal products.

### 5.2. Measuring Polyphenol Intake through Dietary Assessment

Dietary assessment tools greatly affect estimates of polyphenol consumption apart from the limited availability of food composition data. To obtain valid estimates, it is important to collect individual-level data on all polyphenol-containing cereals and cereal-based products consumed by a given population. The most common methods for estimating polyphenol intake from all food sources in epidemiologic studies are Food Frequency Questionnaires (FFQs), food diaries and 24 h Dietary Recalls (24HDRs) [10,85,86,90]. In general, FFQs are designed for a target population and are composed of a list of the most consumed food items [90], querying the frequency at which they are consumed by the respondents. For a study designed to assess polyphenol intake from specific food sources such as cereals, the FFQ should include a representative subset of items from this group, since generic FFQs may not provide reliable estimates of all polyphenol-containing cereal foods. FFQs are commonly used in large epidemiological studies and attempt to capture individuals’ usual consumption, since they rely on a long recall period (one month or one year) to capture the individuals’ typical diet. Food processing, cooking and other culinary treatments known to affect the polyphenolic profile of foods are not usually recorded as the FFQ is a closed-section method, unless cooked and processed variants of items are included. Compared with other methods, the FFQ is easy to self-administer and requires less effort from the respondent, making it more acceptable. On the other hand, it leads to less accurate estimates of dietary intake. Food diaries and 24HDRs are open-ended section methods that generally lead to a more accurate dietary assessment through the recording of all foods and beverages consumed during one day by individuals, including portions or quantities, and detailed descriptions of items: the preservation method, culinary treatment, processing and cooking, place and time of meal. Other food attributes such as integral/refined, fortification, fat/sugar content, and target consumer, can be recorded. Brand names are often registered for commercial foods such as breakfast cereals or infant foods. Accurate estimates of composite foods and beverages that contain one or more polyphenol-containing cereal products, for example, bread, pasta, cookies, and cakes made with mixed flours, are obtained with these methods.

Food records (diaries) impose a high respondent’s burden and are generally affected by high non-response bias. The 24HDR method is less burdensome, and results in higher participation rates, although it has been shown to be less accurate than food records in measuring actual consumption. At the population level, the usual intake is measured through repeated food diaries or 24HDRs. At least two non-consecutive days per participant are recommended [91] because of their independence, so that the information collected is likely to provide a better estimate of intra-individual variability than data collection on consecutive days. The total number of survey days should be spread over one-year and four seasons [91].

Although food diaries and 24HDRs are more accurate than FFQs in estimating actual food intake, they may lead to inaccurate estimates of less frequently eaten foods. To overcome this limitation, information on the frequency of consumption of these foods should be collected with an additional food frequency questionnaire covering all seasons, and this information should be used as a covariate in the estimation of usual intake.

Most of the dietary assessment methods used in epidemiological studies, mainly FFQs, have not been validated to estimate polyphenol intake [92]. Recently, an association between polyphenol intake and urinary phenolic metabolites has been investigated [93].

Xu et al. [93] developed a (poly)phenol-rich diet score for the UK population to estimate the intake of polyphenol of 20 plant-based foods through an FFQ. The association between the diet score and a comprehensive panel of (poly)phenol metabolite levels in 24 h urine was explored. Also, Pounis et al. [27] suggest a PAC score to assess the dietary intake of polyphenol.

### 5.3. Food Consumption Data Sources, Food Categorization and Linkage with Databases on Polyphenol Content

Substantial sources of information of potential use for polyphenol intake assessment at the European level are the EFSA Comprehensive Food Consumption Database (CFCD) [16] and the FAO/WHO GIFT platform [17]. The EFSA CFCD pools consumption data collected through national representative dietary surveys across the European Union, and has a relevant role for the assessment of nutrient intakes of the EU population (see Table 1). For a better harmonization of data across countries, EFSA launched the EU-Menu program aimed at supporting member states in the collection of individual food consumption data using the same recommended methodology (food diaries and 24HDRs). To date, the CFCD has collected 16 EU-Menu surveys on the children population (3 months–9 years) and 20 surveys on adolescents, adults, and the elderly population (10–74 years). The FAO/WHO GIFT is an open-access repository providing individual dietary intake data from over 50 surveys conducted worldwide, although it focuses on low- and middle-income countries. To date, additional datasets are also in preparation, and numerous surveys have been identified as potentially suitable for sharing through the platform, for a total of more than 100 countries in all world regions [17].

An optimal link between food consumption data and occurrence data for the assessment of exposure to compounds in foods must be established. In fact, consumed foods and foods in composition tables are often categorized differently. Food classification is crucial to enable the processing of food consumption data, as it is not possible to interpret dietary data and obtain robust food consumption estimates without the aggregation of individual foods into food groups, either before or after the data collection. However, for an optimal link between food consumption and food composition data, it is necessary to define the foods at a sufficient level of detail. Food processing should also be considered. Matching food as consumed with polyphenol data is facilitated if the same food classification is adopted and/or when food definition detail is similar. Decisions must be made when there are similar rather than exact matches, multiple matches, or no matches (see, for example, Vingrys et al. [10]). For instance, specific polyphenol content can be imputed from similar although different foods (e.g., boiled rice to represent different rice-based products) or the most appropriate item can be selected in case of multiple matches [10]. As shown in Table 1, limited data on polyphenols are available for cereal-based composite foods, such as different types of bread, crispbread and pasta, biscuits and fine bakery products, for which consumption data are reported. In the case of unavailable data for composite foods, these must be disaggregated into single ingredients based on standard recipes or the manufacturers’ information, deriving the proportion of polyphenol-containing food(s) and calculating the polyphenol content of each ingredient. A lack of detail may have an impact on the consumption data for composite foods, especially when FFQs are used, and assumptions must be made about the breakdown of the ingredients. The use of harmonized food classifications and descriptions would improve the linking of databases. In addition, referring to multiple sources of polyphenol data increases the accuracy of estimates and reduces imprecise matching or missing data [10]. A better comparability of results from different studies would also be achieved. All foods, beverages and dietary supplements included in the CFCD and the FAO/WHO GIFT repository are classified according to the FoodEx2, a standardized food classification system based on an extended main list of entries (the FoodEx2 basic codes) organized into groups based on a parent–child hierarchical structure [94]. A catalogue of twenty-eight groups of descriptors (the “facets”) allows the addition of additional characteristics of the basic codes, such as processing (e.g., baking, boiling, dehydrating), packaging format and material, fortification with substances (e.g., vitamins, minerals) and qualitative information (e.g., fat/sugar-related information, integral/refined). The main group “Grains and grain-based products” (Level 1 of the hierarchical structure) consists of around 350 reportable basic FoodEx2 codes organized in main sub-groups (Level 2): “Cereals and cereal primary derivatives”; “Bread and similar products”; “Pasta, doughs and similar products”; “Fine bakery wares”, and “Breakfast cereals”. For example, the “Cereals and cereal primary derivatives” group includes basic terms for any type of cereals and cereal-like grains, milling products and other primary derivatives.

Of the basic codes or terms, 27 are at Level 3 of the hierarchy (e.g., A0BY0 leavened bread and similar; A00EJ muesli and similar mixed breakfast cereals), 116 are at Level 4 (e.g., A005F rye-only bread and rolls; A00EK muesli plain) 167 at Level 5 (e.g., A005H rye bread and rolls, wholemeal; A00DH oat rolled grains), 40 at Level 6 (e.g., A005P rye-wheat bread and rolls, wholemeal; A00DJ rolled oats, instant). See Table 1 for details on FoodEx2 hierarchy levels and codes for the main cereal products as sources of polyphenols.

Information on the level of refinement is either explicated in the basic term (e.g., “A003K Rye flour refined”; “A004B wheat wholemeal flour”) or can be added using a facet descriptor (e.g., A002L#F10.A06HR = barley flour, QUALITATIVE-INFO = integral/not refined). A feature of the FoodEx2 system is its flexibility. This means that basic terms and additional descriptors can be added (or removed) as new analysis and study needs arise. Except for red rice (“A001H rice grain, red”) and red oat (“A000H oat grain, red”), which are used for hay and intended for animal feed and not for human consumption, pigmented cereals are not included among the basic terms of FoodEx2. It should be noted, however, that pigmented cereals are still a niche market in most European countries, and it could be assumed that the number of consumers is still too low to obtain robust consumption from current dietary surveys. Ad hoc planned surveys should be carried out, targeting specific geographical areas and/or sub-populations where these products are known to be consumed more frequently. Another example is wholegrain consumption. In some populations, such as Italy, consumption of wholegrains and wholemeal products is quite low, both in terms of the number of consumers and the quantities consumed. This limits the robustness of the estimates, especially in sub-groups of the population (e.g., different age groups), unless the methods used in the dietary surveys allow a detailed recording of the products consumed [95]. In other European countries, wholegrain products are more consumed, for example, wholemeal wheat bread and rolls is largely consumed by the adult population in The Netherlands, Ireland and Belgium (the amount ranges between 66 and 39 g/die); also, wholemeal rye bread, is prevalently consumed in Latvia, Finland, Denmark and Sweden, with an average amount between 37 and 25 g/die (more than 50% of consumers). Polyphenol exposure has been assessed in numerous epidemiologic studies by using food composition databases. Because of the variability of methods for the evaluation and quantification of polyphenol intake, as discussed previously, and the limitations of the data used to estimate polyphenol exposure in cereals, it makes it difficult to clearly state recommendations on intake [96].

## 6. Conclusions and Future Perspective

-The inclusion of new consumer trends regarding novel cereal-based products, such as nutrient-rich foods, pigmented cereals, dietary supplements, etc., should be reflected in a higher resolution of food compositional data.-Information about the food composition of cereal-based products should consider all cereal species, the geographical and agricultural practices, and all peculiar preparation and cooking procedures that could influence the profiles of polyphenols.-A more frequent update of the scientific literature about the data of cereal composition in the databases could provide research insights on attributes that influence the variability of classic and emerging bioactive compounds of public health importance.-The use of standardized procedures in analytical methods for food composition databases could allow the obtainment of a reliable database for epidemiological studies.-Measuring polyphenol intake requires an optimal link between the consumption and occurrence data of individual food sources, considering food preparation and processing. This could be hampered by differences in the level of detail of food databases.-Dietary assessment tools should be validated for measuring polyphenol intake before use in epidemiologic studies.-Standardized food categorization systems would reduce the subjectiveness of data matching procedures, thereby improving the robustness of estimates.-The use of harmonized food consumption data will improve the comparability of results across population groups; targeted dietary data collection in specific geographical areas/sub-groups should complement the existing information.-The overcoming of the evidenced limitations will certainly be a step towards the formulation of dietary recommendations, which will have a major impact on the health of the population.

## Figures and Tables

**Figure 1 nutrients-16-02155-f001:**
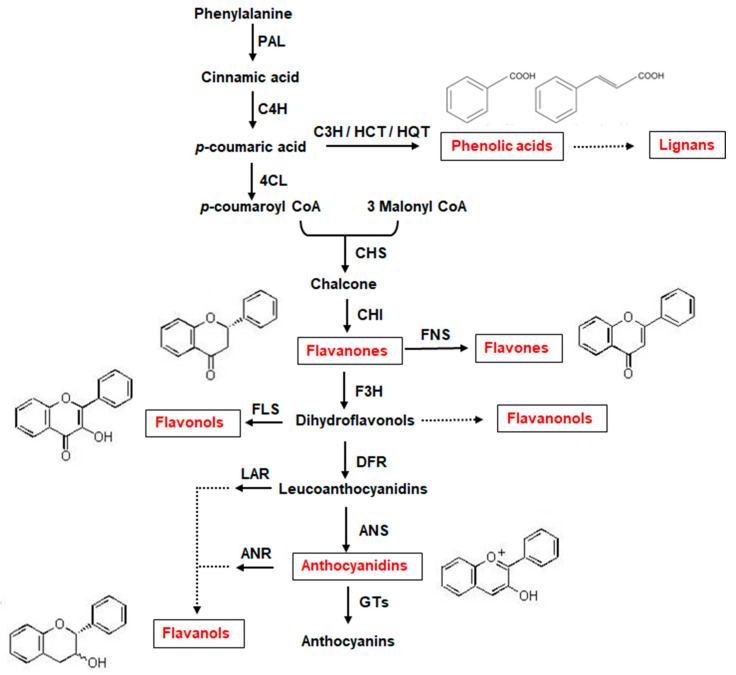
Schematic representation of the polyphenol biosynthetic pathway, with the main classes of polyphenols. PAL, phenylalanine ammonia lyase; C4H, cinnamic acid 4-hydroxylase; C3H, p-coumarate 3-hydroxylase; HCT, hydroxycinnamoyl CoA shikimate/quinate hydroxycinnamoyl transferase; HQT, hydroxycinnamoyl CoA quinate hydroxycinnamoyl transferase; 4CL, 4-coumarate coA ligase; CHS, chalcone synthase; CHI, chalcone isomerase; FNS, flavone synthase; F3H, flavanone 3-hydroxylase; FLS, flavonol synthase; DFR, dihydroflavonol reductase; LAR, leucoanthocyanidin reductase; ANS, anthocyanidin synthase; ANR, anthocyanidin reductase; GTs, glycosyltransferases.

**Table 1 nutrients-16-02155-t001:** Polyphenol sources in cereals and cereal products from different databases and availability of individual dietary intake data in the EFSA *^a^* Comprehensive Food Consumption Database (CFCD), by FoodEx2 code and Exposure Hierarchy level of foods.

Cereals and Cereal Products	Polyphenol Class	FoodEx2 *^b^* Code	FoodEx2 *^b^* Code Description	FoodEx2 *^b^* Hierarchy Level	Polyphenol Data Source	Number of Surveys in CFCD *^c^* (n Countries Covered)
Cereal grains and flour						
Wheat (*Triticum aestivum* L. ssp. *aestivum*), wholegrain flour	Flavones, hydroxycinnamic acids, lignans anthocyanidins	A004B	Wheat wholemeal flour	L5	Phenol-Explorer ePlantLIBRA/eBASIS *^§^*/USDA Database 3.1	27 (14)
Wheat (*Triticum aestivum* L. ssp. *aestivum*), bran	Isoflavones, lignans	A004P	Wheat bran	L4	ePlantLIBRA/eBASIS *^§^*	42 (18)
Wheat (*Triticum aestivum* L. ssp. *aestivum*), refined flour	Flavones, hydroxybenzoic acids, hydroxycinnamic acids, lignans	A003Y	Wheat flour white	L5	Phenol-Explorer	53 (25)
Hard wheat (*Triticum durum* Desf.), wholegrain flour	Hydroxybenzoic acids, hydroxycinnamic acids isoflavones, lignans	A004C	Wheat flour, durum	L5	Phenol-Explorer ePlantLIBRA/eBASIS *^§^*	10 (5)
Hard wheat (*Triticum durum* Desf.), semolina	Hydroxycinnamic acids, lignans	A004F	Wheat semolina	L4	Phenol-Explorer	54 (24)
Maize (*Zea mays* L.), wholegrain	Anthocyanins, hydroxybenzoic acids, hydroxycinnamic acids, lignans	A000T	Maize grain	L5	Phenol-Explorer ePlantLIBRA/eBASIS *^§^*	26 (12)
Maize (*Zea mays* L.), refined flour	Hydroxybenzoic acids, hydroxycinnamic acids, hydroxyphenylacetic acids	A002Q	Maize flour	L5	Phenol-Explorer	47 (24)
Barley (*Hordeum vulgare* L. ssp. *vulgare)*, wholegrain flour	Flavanols, lignans, isoflavones, flavan-3-ols	A002L	Barley flour	L4	Phenol-Explorer ePlantLIBRA/eBASIS *^§^* USDA Database 3.1	12 (6)
Oat (*Avena sativa* L.), wholegrain flour	Hydroxycinnamic acids, lignans isoflavones	A002Y	Oat flour	L4	Phenol-Explorer ePlantLIBRA/eBASIS *^§^*	16 (11)
Oat (*Avena sativa* L.), refined flour	Hydroxybenzoic acids, hydroxycinnamic acids, hydroxyphenylacetic acids	A002Y	Oat flour	L4	Phenol-Explorer	16 (11)
Rice (*Oryza sativa* L.) wholegrain	Hydroxycinnamic acids, isoflavones, lignans, flavan-3-ols	A001D	Rice grain	L5	Phenol-Explorer ePlantLIBRA/eBASIS *^§^* USDA Database 3.1	74 (27)
Rice (*Oryza sativa* L.) parboiled/refined	Hydroxybenzoic acids, hydroxycinnamic acids	A003D; A003E	Rice grain, polished; Rice grain, parboiled	L6; L6	Phenol-Explorer	48 (21); 6 (5)
Rye (*Secale cereale* L.), wholegrain flour	Hydroxybenzoic acids, hydroxycinnamic acids, lignans	A003M	Rye flour, wholemeal	L5	Phenol-Explorer ePlantLIBRA/eBASIS *^§^*	19 (8)
Rye (*Secale cereale* L.), refined flour	Hydroxybenzoic acids, hydroxycinnamic acids, hydroxybenzaldehydes	A003K	Rye flour, refined	L5	Phenol-Explorer	6 (1)
Sorghum (*Sorghum bicolor* L.), wholegrain	Flavanols (proanthocyanidins) Flavones, lignans, cinnamic acid derivatives flavanones	A001L	Sorghum grain	L5	Phenol-Explorer ePlantLIBRA/eBASIS *^§^* USDA Database 3.1	0
Bread						
Bread, wheat, wholegrain flour	Lignans	A005E	Wheat bread and rolls, brown or wholemeal	L5	Phenol-Explorer	71 (26)
Bread, wheat, refined flour	Lignans	A004Y	Wheat bread and rolls, white (refined flour)	L5	Phenol-Explorer	74 (27)
Bread, rye, wholegrain flour	Lignans, phenolic acids	A005H	Rye bread and rolls, wholemeal	L5	Phenol-Explorer	47 (20)
Breakfast cereals						
Breakfast cereals, bran	Lignans	A00ED	Wheat bran rolled flakes	L6	Phenol-Explorer	25 (13)
Breakfast cereals, corn	Lignans	A00DD	Processed maize-based flakes	L5	Phenol-Explorer	71 (27)
Breakfast cereals, muesli	Lignans	A00EJ; A00EK; A00EL	Muesli and similar mixed breakfast cereals; muesli plain; mixed breakfast cereals	L3; L4; L4	Phenol-Explorer	37 (19); 36 (18); 50 (22)
Breakfast cereals, oat, wholemeal	Lignans	A00DL	Oat rolled grains, wholemeal	L6	Phenol-Explorer	9 (6)
Pasta						
Pasta, wholegrain Macaroni, cooked	Lignans	A04LC	Pasta wholemeal	L5	Phenol-Explorer	39 (18)
	Flavan-3-ols	A007P; A007X; A008A	Dried durum pasta; fresh stuffed durum pasta; dried stuffed durum pasta	L6; L6; L6	USDA Database 3.1	41 (21); 4 (3); 3 (2)

*^a^* European Food Safety Authority; *^b^* FoodEx2 Catalogue Browser, Version 1.2.14, Copyright © 2012–2019 EFSA, European Union Public Licence V. 1.2. *^c^* European Comprehensive Food Consumption Database; ^§^ BioActive Substances in Food Information System (2017) eBasis.eurofir.org. Available at: https://ebasis.eurofir.org/Default.asp (Accessed: 24 June 2024).

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
