# Peer review of "Polyphenols in Cereals: State of the Art of Available Information and Its Potential Use in Epidemiological Studies"

_nutrients, 2024, doi:10.3390/nu16132155_

Round 1

Reviewer 1 Report

Comments and Suggestions for Authors

This is an interesting review article on a type of crucial metabolites in the diet, however you need to make some adjustments such as:

Keyword: you must place words that are not repeated in the title. I suggest removing: Polyphenols, Epidemiological and replacing with others.

In the introduction it is interesting to place data on the consumption of this type of food by countries worldwide.

At the end of the introduction, it is necessary to introduce a section explaining the search criteria used to create the article. Types of databases used, search criteria, years in which I am going to narrow the search.

The authors should highlight a section with the importance of this type of biomolecules, how this type of secondary metabolites are formed, classification of families of phenolic compounds, structures, since the authors have not considered it.

Table 1 is very general, since it talks about foods and class of phenols. From Table 1, a second table can be generated in which specific phenolic compounds are classified within each family since the authors only talk about families and not compounds.

The conclusions are clear and concise and the bibliographic references are correctly cited.

Comments on the Quality of English Language

 Minor editing of English language required

Author Response

This is an interesting review article on a type of crucial metabolites in the diet, however you need to make some adjustments.

Our reply: We thank the Reviewer for the exhaustive overview of the manuscript. The manuscript has been improved following his/her suggestions, with the indications of our modifications also given in our replies.

Keyword: you must place words that are not repeated in the title. I suggest removing: Polyphenols, Epidemiological and replacing with others.

Our reply: We thank the reviewer for this suggestion. The keywords “Polyphenols and Epidemiological” have been replaced with “Antioxidants and Observational studies”. Moreover, ‘polyphenol database’ has been changed into ‘composition database’.

In the introduction it is interesting to place data on the consumption of this type of food by countries worldwide.

Our reply: According to reviewer suggestion, the text has been revised as follows “Total world cereal production in 2022 was estimated at 3059 million tonnes. Estimates of domestic supplies for 2021 were highest in Asia (1663.2), followed by North America (393.32), Africa (316.38) and European Union (281.10) (https://www.fao.org/faostat/). Since 2010, there has been a trend towards an increase in these figures. In the European Union, wheat and its products account for the largest share of the cereals available for domestic consumption (115.66 million tonnes), followed by maize (66.11) and barley (46.26) (https://www.fao.org/faostat/)”.

At the end of the introduction, it is necessary to introduce a section explaining the search criteria used to create the article. Types of databases used, search criteria, years in which I am going to narrow the search. 

Our reply: We thank the Reviewer. The search criteria have been reported: “As for food consumption data, the EFSA European Comprehensive Food Consumption Database (CFCD) [16], and the FAO/WHO Global Individual Food Consumption Data Tool (FAO/WHO GIFT) [17] were explored. The former provides data from most European countries, while the latter collects dietary data from all regions of the world, focusing on low- and middle-income countries. The state-of-the-art of databases is reported at the date of the drafting of the present manuscript (May 2024)”.

The authors should highlight a section with the importance of this type of biomolecules, how this type of secondary metabolites are formed, classification of families of phenolic compounds, structures, since the authors have not considered it.

Our reply:  We have added a new paragraph entitled “Defining polyphenols: structure and properties” (Now paragraph 2). Also, a figure (Figure 1) has been included in the manuscript.

Table 1 is very general, since it talks about foods and class of phenols. From Table 1, a second table can be generated in which specific phenolic compounds are classified within each family since the authors only talk about families and not compounds.

Our reply: We thank the Reviewer for this suggestion. We have revised the Table 1, including the availability of individual dietary intake data at European level. Furthermore, a second Table, named S1, has been produced, containing the details of the phenolic compounds, within each class, for cereal grains and cereal-based products.

The conclusions are clear and concise and the bibliographic references are correctly cited.

Our reply: We thank the Reviewer for his/her positive comment.

Minor editing of English language required

Our reply: The paper has been revised to improve the English language.

Reviewer 2 Report

Comments and Suggestions for Authors

Title: Polyphenols in cereals: state of the art of available information and its potential use in epidemiological studies

The manuscript provides a comprehensive overview of the polyphenolic content in cereals and their potential health benefits, emphasizing the gaps in current data and the challenges in assessing polyphenol intake through epidemiological studies. While the topic is interesting, there are several areas that require attention to improve the clarity, comprehensiveness, and scientific rigor of the manuscript.

·      While the introduction provides a solid foundation, it would benefit from a more detailed review of recent literature, especially studies published in the last five years.

·      The manuscript mentions databases like EuroFIR, USDA, and PhenolExplorer but should provide a more indepth comparison of these resources. What specific types of polyphenols are covered in each? How do they compare in terms of data comprehensiveness and update frequency?

·      While the manuscript identifies missing data for pigmented cereals, it should also discuss the potential reasons for these gaps and suggest specific strategies for addressing them.

·      Section 3 is informative but could be more comprehensive. The discussion on the health benefits of pigmented cereals should be supported by more recent and diverse studies.

·      A comparative analysis of the polyphenol content in different pigmented cereals versus nonpigmented varieties would add depth to this section.

·      The manuscript mentions the challenges in estimating polyphenol intake but lacks a detailed discussion on the methodological limitations of existing dietary assessment tools. A comparison of the accuracy and reliability of FFQs, food diaries, and 24hour recalls in this context would be beneficial.

·      Identifying gaps in current knowledge and suggesting directions for future research would enhance the conclusion’s impact.

·      The manuscript mentions the limitations of existing databases but does not adequately address the potential impact of these limitations on epidemiological research. A more detailed analysis of how these gaps affect study outcomes would be valuable.

·      The manuscript highlights the need for standardized procedures but does not provide concrete examples or proposals for how such standardization could be achieved.

·      The inconsistencies in food categorization and data collection methods across different studies are mentioned but not explored in detail. This section could benefit from examples of how these inconsistencies have impacted previous research findings.

·      There is a lack of discussion on regional variability in cereal consumption and polyphenol content. Including data or studies from different geographical regions would provide a more comprehensive understanding.

·      The manuscript briefly mentions bioavailability but does not delve into the factors affecting polyphenol bioavailability from cereals. This is a critical aspect that should be addressed, as it influences the actual health benefits derived from consumption.

 Technical and Minor Issues:

Some sections of the manuscript are dense and could be made clearer with the use of subheadings and more concise language.

Ensure consistent use of terminology throughout the manuscript, particularly when referring to polyphenols and their various subclasses.

Author Response

The manuscript provides a comprehensive overview of the polyphenolic content in cereals and their potential health benefits, emphasizing the gaps in current data and the challenges in assessing polyphenol intake through epidemiological studies. While the topic is interesting, there are several areas that require attention to improve the clarity, comprehensiveness, and scientific rigor of the manuscript.

Our reply: We thank the Reviewer for the appreciation of the topic. We tried our best to improve the manuscript and made requested changes in the manuscript.

While the introduction provides a solid foundation, it would benefit from a more detailed review of recent literature, especially studies published in the last five years.

Our reply: We have checked the references in the Introduction, and we have updated them.

The manuscript mentions databases like EuroFIR, USDA, and PhenolExplorer but should provide a more indepth comparison of these resources. What specific types of polyphenols are covered in each? How do they compare in terms of data comprehensiveness and update frequency?

Our reply: We have revised the text evidencing in yellow the differences among the databases reported in the first part of section 3. In general, the up-to-date food composition database depends on the funding sources within projects dedicated to components of the main food items and is lacking in many areas, particularly in developing countries.

While the manuscript identifies missing data for pigmented cereals, it should also discuss the potential reasons for these gaps and suggest specific strategies for addressing them.

Our reply: We have implemented the text in section 3 as follows: “The food and nutrient databases in the current forms do not provide truly comprehensive food composition data [32]. This is even more evident in the case of cereals and particularly of pigmented cereals. Furthermore, the polyphenol characterization depends on species and cultivars, climate, agronomic practices, post-harvest and food processing, methods of domestic preparation of cereal-based foods. This complexity requires validated and rigorous analytical methods that enable a specific and reliable identification of single types of polyphenols, as evidenced by Schroeter et al. [33] and Yeung [34] for flavanols and procyanidins. Despite certain analytical methods and procedures are widely used (e.g. HPLC, GC-MS), standard analytical methods for the determination of polyphenol composition have not been established for all polyphenols classes. To this aim, accredited analytical standard methods (i.e., validated ICC, AOACI, ISO, etc) should be adopted, in which sample collection and preparation, detection systems, identification are precisely described and standardized. These standardized procedures will overcome those limitations that currently hinder cross-study comparisons, meta-analyses, and multi-source food intake data evaluations. With these technical improvements, the ability to integrate information from different data sources will be possible and will allow to establish a definitive understanding of what is currently known and what is missing in order to make food composition data more complete and provide findability, accessibility, interoperability, and reusability (FAIR)”.

  • Section 3 is informative but could be more comprehensive. The discussion on the health benefits of pigmented cereals should be supported by more recent and diverse studies.

Our reply: Section 3 has been implemented and made more comprehensive by including traditional pigmented rice or corn from Asia or South America, which are being used as medications or to prepare traditional foods: “Scientific studies have attributed several health benefits to pigmented rice, such as antioxidant, antidiabetic and anticancer properties [24]. Some traditional pigmented rice varieties are used for medications in the Ayurveda traditional medicine and as functional food for promoting lactation, such as red rice Rakthashali from Kerala, India, or as beneficial food to ameliorate blood circulation, gastritis and peptic ulcers, such as black rice Kavuni from Tamil Nadu, India [24,58,59]”.  The discussion on the health benefits of pigmented cereals has been also implemented with more recent reviews describing animal in vivo studies as well as with the description of additional and more recent human intervention studies.

  • A comparative analysis of the polyphenol content in different pigmented cereals versus nonpigmented varieties would add depth to this section.

Our reply: We thank the Reviewer for this suggestion. In section 4 a description of polyphenol classes determining pigmentation in the most important cereal species, such as rice, corn and wheat has been included: “Compared with non-pigmented cereals, pigmented cereals contain a high concentration of proanthocyanidins (i.e. red rice), phlobaphenes (i.e. red corn) and anthocyanins in rice, wheat and corn [29,37]. Concerning anthocyanins, black and purple rice contain high amounts of cyanidin 3-glucoside [24], whereas blue and purple corn contain cyanidin 3- glucoside, peonidin 3-glucoside, pelargonidin 3-glucoside and malonylated derivatives in the aleurone or in the pericarp of kernels, respectively [46]. Finally, blue-aleurone wheats contain delphinidin 3-glucoside, delphinidin 3-rutinoside and malvidin 3-glucoside while in purple-pericarp wheat and durum wheat cyanidin 3-glucoside, and/ or peonidin malonylglucoside, peonidin 3-galactoside and malvidin 3-glucoside have been observed [28,30]. These compounds have been associated with high antioxidant capacity and anti-inflammatory activities [53]. Pigmented cereals are treated as suitable ingredients as they provide functional attributes apart from colour (for review see Bassolino et al., [54])”.

  • The manuscript mentions the challenges in estimating polyphenol intake but lacks a detailed discussion on the methodological limitations of existing dietary assessment tools. A comparison of the accuracy and reliability of FFQs, food diaries, and 24hour recalls in this context would be beneficial.

Our reply: We thank the Reviewer for this suggestion. Although main limitations of FFQs in estimating polyphenol intake were already outlined in the manuscript, the discussion was improved adding the following texts in section 5: “Compared with other methods, the FFQ is easy to self-administer and requires less effort from the respondent, making it more acceptable. On the other hand, it leads to less accurate estimates of dietary intake”.

“Food record (diaries) impose a high respondent’s burden and are generally affected by high non-response bias. The 24HDR method is less burdensome, and results in higher participation rates, although it has been shown to be less accurate than food record in measuring actual consumption. At population level the usual intake is measured through repeated food diaries or 24HDRs. At least two non-consecutive days per participant are recommended [91] because of their independence, so that the information collected is likely to provide a better estimate of intra-individual variability than data collection on consecutive days. The total number of survey days should be spread over one-year and four seasons [91].

Although food diaries and 24HDRs are more accurate than FFQs in estimating ac-tual food intake, they may lead to inaccurate estimates of less frequently eaten foods. To overcome this limitation, information on the frequency of consumption of these foods should be collected with an additional food frequency questionnaire covering all seasons, and this information should be used as a covariate in the estimation of usual intake.

Most of the dietary assessment methods used in epidemiological studies, mainly FFQs, have not been validated to estimate polyphenol intake [92]. Recently, an association between polyphenol intake and urinary phenolic metabolites has been investigated [93].

Xu et al. [93] developed (poly)phenol-rich diet score for the UK population to estimate the intake of polyphenol of 20 plant-based foods through a FFQ. The association between the diet score and a comprehensive panel of (poly)phenol metabolite levels in 24 h urine was explored. Also, Pounis et al. [27] suggest the PAC score to assess the dietary intake of polyphenol”.

  • Identifying gaps in current knowledge and suggesting directions for future research would enhance the conclusion’s impact.

Our reply: We thank the Reviewer. The gaps have been identified and discussed throughout the text, particularly in the sections 3 and 5. Also, conclusions and future directions have been checked and revised (section 6).

  • The manuscript mentions the limitations of existing databases but does not adequately address the potential impact of these limitations on epidemiological research. A more detailed analysis of how these gaps affect study outcomes would be valuable.

Our reply: Some impacts of databases limitations on epidemiological research have been discussed in sections 4 and 5. Our reply to the 2 comments below and related text modifications partially address also the present comment. Text has been also added in conclusion of Section 5: Polyphenol exposure has been assessed in numerous epidemiologic studies by using food composition databases. Because of the variability of methods for the evaluation and quantification of polyphenol intake, as discussed previously, and the limitations of data used to estimate polyphenol exposure in cereals makes it difficult to clearly state recommendations on intake [96]”.

  • The manuscript highlights the need for standardized procedures but does not provide concrete examples or proposals for how such standardization could be achieved.

Our reply: We thank the Reviewer for this suggestion. About the standardization of the methods a suggestion has been reported in section 3: “Despite certain analytical methods and procedures are widely used (e.g. HPLC, GC-MS), standard analytical methods for the determination of polyphenol composition have not been established for all polyphenols classes. To this aim, accredited analytical standard methods (i.e., validated ICC, AOACI, ISO, etc) should be adopted, in which sample collection and preparation, detection systems, identification are precisely described and standardized. These standardized procedures will overcome those limitations that currently hinder cross-study comparisons, meta-analyses, and multi-source food intake data evaluations. With these technical improvements, the ability to integrate information from different data sources will be possible and will allow to establish a definitive understanding of what is currently known and what is missing in order to make food composition data more complete and provide findability, accessibility, interoperability, and reusability (FAIR)”.

  • The inconsistencies in food categorization and data collection methods across different studies are mentioned but not explored in detail. This section could benefit from examples of how these inconsistencies have impacted previous research findings.

Our reply: Thank you for this comment. We improved the section adding discussion and some examples on how different food categorizations and descriptions could have an impact in measuring polyphenol intake. Text has been added: Matching food as consumed with polyphenol data is facilitated if the same food classification is adopted and/or when food definition detail is similar. Decisions must be made when there are similar rather than exact matches, multiple matches, or no matches (see for example Vingrys et al. [10]). For instance, specific polyphenol content can be imputed from similar although different foods (e.g., boiled rice to represent different rice-based product) or the most appropriate item can be selected in case of multiple matches [10]. As shown in Table 1, limited data on polyphenols are available for cereal-based composite foods, such as different types of bread, crispbread and pasta, biscuits and fine bakery products, for which consumption data are reported. In case of unavailable data for composite foods, these must be disaggregated into single ingredients based on standard recipes or manufactures’ information, deriving the proportion of polyphenol containing food(s) and calculating the polyphenol content of each ingredient. Lack of detail may have an impact on the consumption data for composite foods, especially when FFQs are used, and assumptions must be made about the breakdown into ingredients”.

  • There is a lack of discussion on regional variability in cereal consumption and polyphenol content. Including data or studies from different geographical regions would provide a more comprehensive understanding.

Our reply: Some more data on cereal consumption in different world geographical regions has been added at the beginning of the Introduction. The mean intake reported 219 g/day is calculated based on the adult population of 21 European countries that implemented the food consumption surveys in the last 10 years. The lowest intake of cereal was observed in Estonia with 1 g/day.  

In addition, a discussion about the studies of pigmented cereal varieties cultivated in different geographical regions and the complexity of differences in their polyphenol composition has been implemented in Section 2: “ Overall, the amount and specific polyphenolic profile of cereal species and varieties may depend on the plant genetic background, the environmental growth conditions, including the possible occurrence of abiotic and biotic stresses, as well as the agronomic practices applied [9]. Furthermore, different varieties of pigmented cereals, adapted to local pedoclimatic conditions, are cultivated all around the world. As an example, pigmented corn is mainly cultivated in Central and South America (i.e. Mexico, Peru and Bolivia), but many other varieties considered promising functional foods are also cultivated in other countries (i.e. Russia, Turkey, Europe, Thailand) [22,23]. The polyphenol composition of many of these varieties have been determined, indicating the existence of specificities in anthocyanin glycosylation and acylation, and of differences in hydroxycinnamic acid composition that contribute to the complexity of polyphenol characterization of these geographical accessions (for a review see Colombo et al. [22]). A similar complexity has been reviewed for pigmented rice and wheat varieties [24,25]”. 

  • The manuscript briefly mentions bioavailability but does not delve into the factors affecting polyphenol bioavailability from cereals. This is a critical aspect that should be addressed, as it influences the actual health benefits derived from consumption.

Our reply: We have implemented Section 3 with a short description of factors affecting polyphenol bioavailability, such as the molecular structure of polyphenols (e.g. the influence of decorations like glycosylations and acylations) and their interaction with intestinal transporters, but also the covalent/non-covalent binding of flavonoids with components of the food matrix and possible technological methods to improve their release. However, it is worth noting that only a very limited number of studies have analysed the influence of cereal food matrix on both bioaccessibility and bioavailability of flavonoids so far and more research will be necessary in this field to understand/improve factors affecting polyphenols bioavailability in cereals.

The following text has been reported “In general, the molecular structure of polyphenols determines their bioavailability, but also the type of food matrix may significantly affect their absorption. As an example, acylated anthocyanins are less bioavailable than non-acylated [37]. Glycosylated anthocyanins are more stable and water soluble than their aglycone counterparts, but their bioavailability results limited since they can only be absorbed through glucose transporters, such as SGLT [38]. The overall bioavailability of 13C-labelled cyanidin 3-glucoside (C3G) has been in fact estimated to be 12%, by measuring plasmatic levels of both C3G, its phase I and phase II metabolites and microbiota degradation products [39,40]. Nonetheless, polyphenols need to be released from the food matrix in order to be absorbed and transported via the bloodstream to the target tissue, in order to display their biological activity. Some studies have highlighted that flavonoids can bind the food matrix through covalent or non-covalent bonds, that they can influence nutrient absorption and in turn be influenced by nutrients in their bioavailability and biological activity (for a review see Zhang et al [41]). As an example, wheat proteins may interact with flavonoids forming indigestible complexes that reduce their antioxidant activity [42]. However, the addition of citric acid in food preparations and supplements in order to avoid flavonoids oxidation was found to enhance their bioavailability by releasing them from the food matrix of pigmented maize [43]. On the other hand, despite thermal treatment of pigmented maize flour reduced to some extent the anthocyanin content, the high temperature applied during cooking processes had the advantage to increase their bioavailability by releasing them from the food matrix [44]. More studies are however needed to understand the influence of cereal as well as other plant food matrices on both bioaccessibility and bioavailability of polyphenols [41]”.

Technical and Minor Issues:

Some sections of the manuscript are dense and could be made clearer with the use of subheadings and more concise language.

Our reply: Thank you for this observation. In order to make reading easier, Section 5 has been reorganized and divided into three subsections. Subheadings were added.

Ensure consistent use of terminology throughout the manuscript, particularly when referring to polyphenols and their various subclasses.

Our reply: We have checked the terminology in the manuscript.

Round 2

Reviewer 1 Report

Comments and Suggestions for Authors

The authors have made the requested corrections and the manuscript can now be considered for publication.